# Effects of acute stress on biological motion perception

Jifu Wang[1], Fang Shi[1], Lin Yu[2] *

1 College of Education and Physical Education, Yangtze University, Jingzhou, China, 2 Neurocognition and Action-Biomechanics Research Group, Faculty of Psychology and Sports Science, Bielefeld University, Bielefeld, Germany

* lin.yu@uni-bielefeld.de

**Data Availability Statement:** Data (i.e., EEG and behavioral raw data) for the current study are available here, https://doi.org/10.4119/unibi/2991696.

## Abstract

Biological motion perception is an essential part of the cognitive process. Stress can affect the cognitive process. The present study explored the intrinsic ERP features of the effects of acute psychological stress on biological motion perception. The results contributed scientific evidence for the adaptive behavior changes under acute stress. After a mental arithmetic task was used to induce stress, the paradigm of point-light displays was used to evaluate biological motion perception. Longer reaction time and lower accuracy were found in the inverted walking condition than in the upright walking condition, which was called the "inversion effect". The P2 peak amplitude and the LPP mean amplitude were significantly higher in the local inverted perception than in the local upright walking condition. Compared to the control condition, the stress condition induced lower RT, shorter P1 peak latency of biological motion perception, lower P2 peak amplitude and LPP mean amplitude, and higher N330 peak amplitude. There was an "inversion effect" in biological motion perception. This effect was related to the structural characteristics of biological motion perception but unrelated to the state of acute psychological stress. Acute psychological stress accelerated the reaction time and enhanced attention control of biological motion perception. Attention resources were used earlier, and less attentional investment was made in the early stage of biological motion perception processing. In the late stage, a continuous weakening of inhibition was shown in the parieto-occipital area.

## Introduction

Biological motion perception refers to the perceptual process of judging motion characteristics and gender, and it is typically assessed through biological light spots [1]. This method provides a valuable index of the perception of complex motion patterns in the visual system [2]. The perceiver's emotional state has been shown to affect biological motion perception [3, 4]. For example, one study found that social anxiety affected the orientation bias of biological movement [3]. Another study found that people judged the speed of biological light spots as significantly faster when in a fear state than in a non-fear state [4]. Research in this area has assessed the influence of emotional state on behavioral measures of biological motion perception.

**Funding:** The present study was supported by Humanities and Social Sciences Projects Funded by the Ministry of Education (Grant No. 23YJCZH206), which funded the language editing services for this manuscript. This study was also supported by the Scientific Research Project of the Hubei Provincial Department of Education (Grant No. Q20221313), which funded the cost of experimental supplies and the compensation for participants. The funders had no role in study design, data collection and analysis, decision to publish, or preparation of the manuscript.

**Competing interests:** NO authors have competing interests.

However, no research has explored the internal mechanism of these effects. In the current study, we examined the effects of acute psychological stress on biological motion perception and attention as a mechanism of these effects.

Acute stress appears to affect perceptual processing that depends on the prefrontal cortex by increasing the secretion of norepinephrine and dopamine [5, 6]. Acute stress can be induced by the improved Montreal Imaging Stress Task (MIST) [7]. Attention and cognitive load are both likely to play a role in the link between acute stress and biological motion perception [8, 9]. For example, acute stress has been shown to promote general cognitive function when there is a low cognitive load or when cognitive tasks are simple [10], and acute stress also improves motion-in-depth perception performance under these conditions [11]. Acute stress may have a facilitative effect by accelerating processing speed and enhancing attention [12], attention control [13], and alertness [14]. The processing of local information from biological motion could have appeared to benefit from pre-attention processing [15]. Therefore, it could be speculated that the positive association between acute psychological stress and biological motion perception is mediated by attention control. In the current study, we tested this possibility by assessing the biological motion perception of stimuli that varied in contour characteristic (global or local) and motion characteristic (upright or inverted).

Johansson used point-light displays (PLD) to study people's ability to recognize biological motion. PLD assesses biological motion perception by providing biological light spot stimulation that carries information about the contour and motion characteristics of the human body [16]. An earlier study using PLD found that people quickly detected the contour and motion characteristic of biological motion when the light spots were dynamically presented, and when the light spots were static, people perceived slower biological movement [17]. When the shape information was fuzzy, the primary basis for people to identify biological motion came from motion information. If the motion information was shielded, people relied on shape information to identify body posture and other social information [17]. In this study, the dynamic diagram of biological light spots was used to evaluate the ability of biological motion perception.

The brain regions involved in biological motion perception include the Superior Temporal Sulcus (STS), Fusiform Gyrus (FG), Lingual Gyrus (LG), and Middle Temporal Area (MT+/V5) [9]. These areas would interact with brain areas responsible for motor planning and would be involved in attention allocation, such as sensorimotor areas and the premotor cortex (PMC) [18]. STS and FG are the critical lobes and are located in the temporal and the occipital temporal lobes, respectively. The post-STS region is mainly responsible for the processing of motion information from light spot stimulation, whereas the FG region is primarily responsible for processing visual information provided by the body [19]. Electroencephalography (EEG) studies have further provided information on the temporal course of brain activation while exposed to PLD, revealing the presence of four event-related potential (ERP) components [20–23]. The first component, P1, is highly associated with the primary feature coding of moving dots pattern, which peaks around 100ms time-locked to the onset of stimulus over the extrastriate cortex [20]. The second component, P2, is one of the indicators to evaluate working memory, which peaks around 200ms (time-locked to the onset of stimulus) over the occipital area [21]. The third component, N330, has been shown to be produced by the processing of perceived biological movement, and it is one of the indicators to evaluate the attentional control of biological motion perception, which peaks around 330ms (time-locked to the onset of stimulus) over the posterior superior temporal sulcus (pSTS) [22]. The fourth component, the late positive potential (LPP), has been suggested to be involved in the process of biological motion perception [23]. Accordingly, based on the above-mentioned findings [20–24], we mainly focused on the ERP components P1, P2, N330, and LPP in this study.

The purpose of this study was to explore attention control as an internal mechanism of the positive effects of acute psychological stress on biological motion perception. The improved MIST task [7] was used to induce acute psychological stress in healthy young adults in the laboratory. The sequence diagram of biological light spots [16] was used to evaluate the biological motion perception. Biological motion perception was assessed using ERP techniques with high temporal resolution. We hypothesized that (a) participants will be faster in processing biological motion perception under the stress vs. control condition, (b) the P1, P2, and N330 ERP components will show shorter latency and higher amplitude under the acute stress vs. control condition, and (c) there will be an "inversion effect" in the process of biological motion perception.

## Methods

### Participants

G*Power 3.1.9 was used to conduct an a priori power analysis [25]. Using a repeated measures design, the sample size needed to detect a medium effect size (d = 0.25) with power = .80 was 16. Based on this information and on the sample size in a previous study [10], our sample size was set as $N = 24$ (12 males). A larger sample was used because this could increase the reliability of the experimental results. The mean age of participants was 19.36 (SD = 1.34) years. All participants had normal or corrected-to-normal vision, and all were right-handed. This experiment was approved by the Human Experiment Ethics Committee of the first author's affiliated university.

The 24 college students who volunteered met the following selection criteria: (1) stable emotional state: score > 25 on the Emotional State Assessment Scale [26]; (2) minimal anxiety and depression: score < 48 on the State-Trait Anxiety Inventory [27] and score < 5 on the Beck Depression Inventory [28]; (3) not in a state of loneliness: score < 48 on the UCLA Loneliness Scale [29]; and (4) good spatial thinking ability: score > 75 on the Mosaic Pattern Test [30]. This relative homogeneity reduces the chance that confounding variables will affect the results. All participants provided written informed consent and received 50 RMB (about 7 USD) after the experiment.

### Design and stimuli

We employed a 2 (contour characteristic: global vs. local) × 2 (motion characteristic: upright vs. inverted) × 2 (stress level: control vs. stress) within-subjects experimental design. The dependent variables were the reaction time and accuracy of biological motion perception, the peak amplitude and peak latency of P1, P2, and N330, and the mean amplitude of the late positive potentials (LPP).

The modified MIST task paradigm was used to induce acute psychological stress [7]. This paradigm presents 200 multiplication problems (e.g., 2.16 × 4.78; the problem list used in this study can be found in S1 Appendix). Participants are asked to judge whether the multiplication result is larger than ten or less than ten. They are instructed to press f on the keyboard if they think the multiplied result is less than ten and to press j otherwise. The modified MIST task has been shown to have a stress-inducing effect [9, 10, 25]. For example, in one study, the task induced significantly higher self-reported state anxiety and lower positive affect than baseline [11].

Moving light dots were used as stimuli in the assessment of biological motion perception [31]. The dots were five arc points in size, white, had a density of 1.85 points/degree$^2$, and had a moving speed of 4.0˚/s. Videos of the light dots on a black background were created using Matlab and Psychtoolbox. All videos were formatted in Windows Media Video (WMV), and

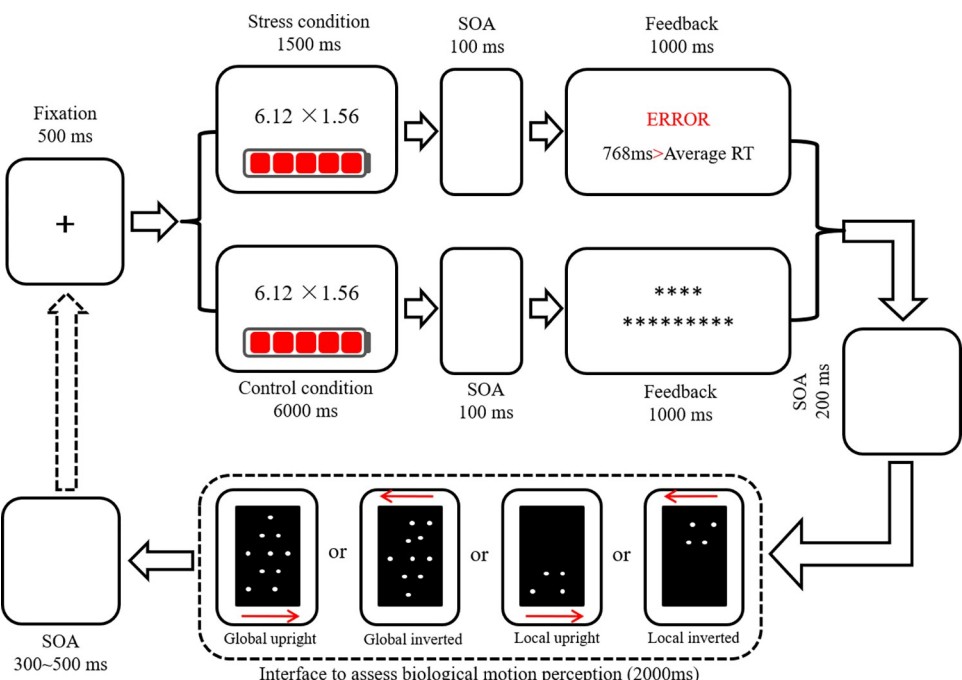

**Fig 1. A flow chart of the experimental task was used to test the effects of acute psychological stress on biological motion perception.** In the formal experiment, the red arrow will not be marked in the detection interface of biological motion perception.

the resolution was set at $1024 \times 768$. The videos were displayed on a 19-inch monitor about 70 cm in front of the participants.

Assessments were made of both global and local biological motion perception. Global biological motion perception was tested using 12 dots representing the main joints of the body, and the dots simulated the movement of the human body walking left or right. Local biological motion perception was tested using four dots representing the foot and the knee joint. Earlier research found that foot movement played a crucial role in the inversion effect of local motion [1, 32].

In order to avoid the effects of different visual angles and the number of dots on the experimental results, a black background was added to present the global and local biological motion dots [33] (see Fig 1). The height of the black background was 115 cm, and the width was 65 cm. The 4 or 12 light dots simulated the movement of the human body walking left or right in the black rectangle. Both global motion and local motion included upright walking and inverted walking, specifically upright left or right walking, inverted left walking, or right walking.

## Procedure

**Preparation.** The experimenter explained the experimental requirements and precautions to the subjects. Subjects completed the informed consent form, the four self-report scales, and the handedness questionnaire. During the experiment, the subjects were asked to look ahead, remain 70 cm from the computer screen, and try not to shake their body. To adjust the breathing rhythm, participants were then asked to look at a relaxing picture presented on the screen and imagine themselves immersed in the relaxed scene.

**Practice and formal experiment.** Feedback was provided in the practice trials but not in the formal experiment. Under the stress condition, the feedback interface presented the

reaction time (RT) and a comparison between the subject's reaction time (RT) and that of the average person. The RTs of the average person were fabricated based on about 700 random numbers. Under the control condition, a set of asterisks was presented in the feedback interface. The mental arithmetic items in the practice experiment (10 trials) were different from those in the formal experiment. The formal experiment included two blocks: the control condition (200 trials) and the stress condition (200 trials). Global upright, global inverted, local upright, and local inverted walking each contained 50 trials. In order to eliminate an item effect, the same mental arithmetic items were used in the control and stress conditions.

The interval between the stress and control conditions was 10 minutes. In the global biological motion task, there were 12 light dots showing images similar to human motion. The 12 light dots simulated walking left and right when the moving figure was upright or inverted (upside down). In the local biological motion task, there were four light dots showing images similar to leg motion. The four light dots simulated walking left or right when the moving figure was upright or inverted (upside down). The subjects were instructed, "If the light dots on the screen are walking left, please press f on the keyboard with your left index finger. If the light dots are walking to the right, please press j on the keyboard with your right index finger."

### Behavioral data recording and analysis

Behavioral data (RT and percent accuracy) were collected by E-prime 2.0. SPSS 17.0 software generated descriptive statistics for these variables under the control and stress conditions. A 2 (contour characteristic: global vs. local) × 2 (motion characteristic: upright vs. inverted) × 2 (stress level: control vs. stress) repeated measures analysis of variance (RM-ANOVA) was used to test differences in RT across conditions. If a main effect was significant, the LSD post hoc test was performed. If an interaction was significant, a simple effect analysis was performed. Differences across conditions in percent accuracy were tested with chi-square analysis.

### Electrophysiological recording and analysis

Brain activations were recorded using a 64-channel EEG amplifier with Brain Vision Recorder software (Brain Products, Munich, Germany). According to the international 10–10 system, 64 electrodes were placed on an elastic cap. The impedance of all electrode sites was under five kΩ. The electrode FCz was used as the recording reference, and the electrode AFz was selected as the recording ground. Two extra electrodes were used to record horizontal and vertical electrooculography (EOG). All signals were sampled at 1000 Hz before digitization and filtered from 0.01 to 100 Hz online.

EEG raw signals were processed offline with the Brain Vision Analyzer software (version 2.0). The range of the IIR filter was 0.01 to 35 Hz in the processing of EEG data offline. Independent component analysis (ICA) was applied for EOG correction [34]. The semiautomatic mode of ocular correction was used to eliminate the EOG component in the blink interval. The length of the ICA interval was 50 s. The duration of EEG superposition was the 1200 ms epoch (time-locked to the onset of the light dots), and the period of the first 200 ms was used for baseline correction. Trials with artifact rejection (peak-to-peak amplitude larger than 80 μV) were removed from the grand average. The average number of superpositions in each trial was 42. This number met the minimum requirement of 30 proposed in a previous study [35].

The ERP components' electrode sites and time windows were selected based on the grand average of ERPs and the results of previous studies [36, 37]. The PO3, O1, POz, O2, and PO4 (70–140 ms) sites were selected for the P1 peak latency and amplitude. The P3, P5, P4, and P6 (150–250 ms) sites were selected for the P2 peak latency and amplitude. The Fz and F1 (200–

350 ms) sites were selected for the N330 peak latency and amplitude. The CPz, CP1, and CP2 (400–800 ms) sites were chosen for LPP. According to the time window of each component, the peak latency and amplitude were measured by baseline-peak. Main effects were tested by repeated measures analysis of variance, and simple effects analyses were used to interpret any interaction effects. The Greenhouse–Geisser correction was used to adjust for sphericity violations when there was a factor with more than two levels.

## Results

### Stress manipulation check

Previous studies found that the improved MIST task can induce acute psychological stress, showing lower positive psychological arousal levels and higher state anxiety [11, 38]. In the mental arithmetic task, the results showed that the RT was longer in the control condition versus the stress condition, t = -8.19, $p < .01$. In terms of accuracy, the control condition was significantly higher than the stress condition ($\chi^2 = 20.38$, $p < .01$). Combined with the previous results [11, 13, 39], our results suggest that the MIST task effectively induced stress.

### Behavioral data

**Reaction time.** In terms of biological motion perception tasks, the main effect of stress level was significant, $F(1,23) = 9.78$, $p < .01$, $\eta_p^2 = .30$, $M_{stress} < M_{control}$, see **Fig 2**. The main effect of contour characteristic was not significant, $F(1,23) = 1.36$, $p > .05$, $\eta_p^2 = .06$. The main effect of motion characteristic was significant, $F(1,23) = 161.49$, $p < .01$, $\eta_p^2 = .88$, $M_{upright} < M_{inverted}$. The interaction effect between the contour characteristic and motion characteristic was significant, $F(1,23) = 17.47$, $p < .01$, $\eta_p^2 = .43$. Simple effects analyses showed that the RT of global motion was significantly shorter than that of the local motion in the upright walking condition ($p < .01$). RT of upright walking was significantly shorter than that of inverted walking in global or local motion conditions. There was no significant difference in the two-way interaction of the three independent variables ($p > .05$). The reaction time for each condition can be seen in **Fig 2**.

**Accuracy.** A chi-square test was conducted to detect accuracy differences across stress levels, motion characteristics, and contour characteristics. There was no significant difference between the control condition and stress condition, $\chi^2 = 0.29$, df = 1, $p > .05$. There was a

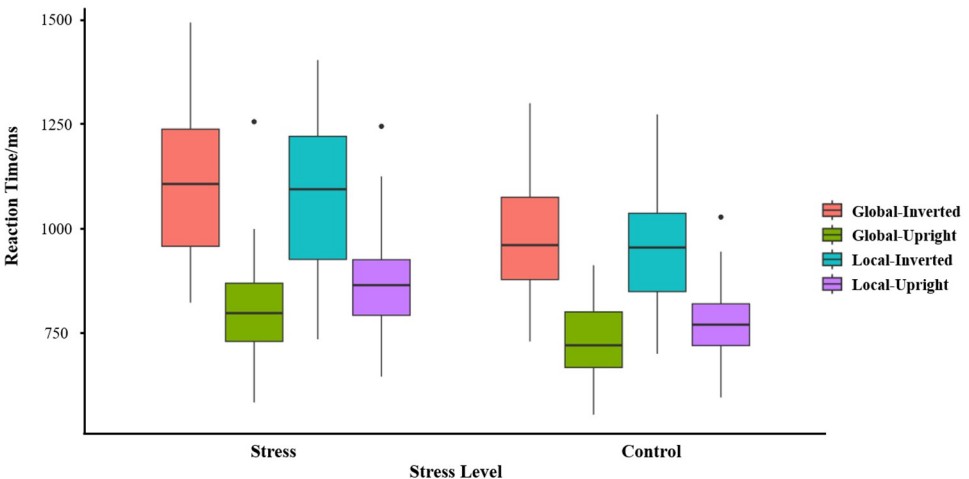

**Fig 2. Reaction time for different experimental conditions.**

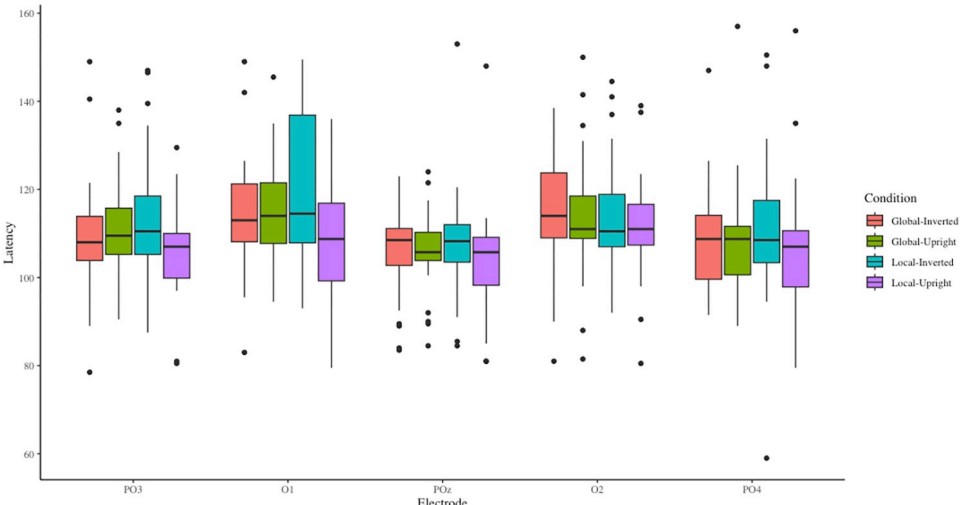

**Fig 3. P1 peak latency for different contour and motion characteristics over the selected electrodes.** Please note that the latency data illustrated here is the average of the stress and control conditions (because the effect of stress level was not significant).

significant effect between the inverted walking and upright walking condition, $\chi^2 = 29.85$, df = 1, $p < .01$, $M_{upright} > M_{inverted}$. There was no significant difference between the global and local motion condition, $\chi^2 = 3.06$, df = 1, $p > .05$. There was no significant difference in the two-way interaction of the three independent variables ($p > .05$).

## Electrophysiological data

**P1 peak latency.** The main effect of stress level was not significant, $F(1,23) = 3.13$, $p = .09$, $\eta_p^2 = .12$, $M_{stress} = 109.82$ms, $M_{control} = 111.16$ms. The main effect of the contour characteristic was not significant, $F(1,23) = 0.19$, $p > .05$, $\eta_p^2 = .01$. The main effect of motion characteristic was significant, $F(1,23) = 8.66$, $p < .01$, $\eta_p^2 = .27$, $M_{upright} = 111.76$ ms, $M_{inverted} = 109.22$ ms. Post hoc testing showed that the P1 peak latency of inverted walking was significantly longer than that of upright walking ($p < .01$), and this P1 peak latency was the average of PO3, O1, POz, O2, and PO4 sites. The main effect of the electrode site was significant, $F(2.94,67.75) = 7.39$, $p < .01$, $\eta_p^2 = .24$. Post hoc testing showed that P1 peak latency was significantly shorter at POz than at O1 and O2 electrode sites ($p < .01$), and the other pairwise comparisons were not significant ($p > .05$).

The interaction effect between the contour characteristic and motion characteristic was significant, $F(1, 23) = 7.93$, $p = .01$, $\eta_p^2 = .26$. Simple effects analyses showed that the P1 peak latency of inverted walking was significantly longer than that of upright walking in local motion condition ($p < .01$). The P1 peak latency of local motion was shorter than that of global motion in upright walking condition ($p < .05$). The interaction effect of electrode site, contour characteristic and motion characteristic was significant, $F(1, 23) = 3.40$, $p = .01$, $\eta_p^2 = .13$. Simple effects analyses showed that P1 peak latency of inverted walking was significantly longer than that of the upright walking in the PO3 or O1 sites and local motion conditions ($p < .01$). The P1 peak latency of local motion was significantly shorter than that of the global motion in the PO3 or O1 sites and upright walking condition ($p < .05$). The P1 latency induced of local motion was significantly longer than that of global motion in the O1 site and inverted walking condition ($p < .05$). The P1 latency for each condition can be found in **Fig 3**.

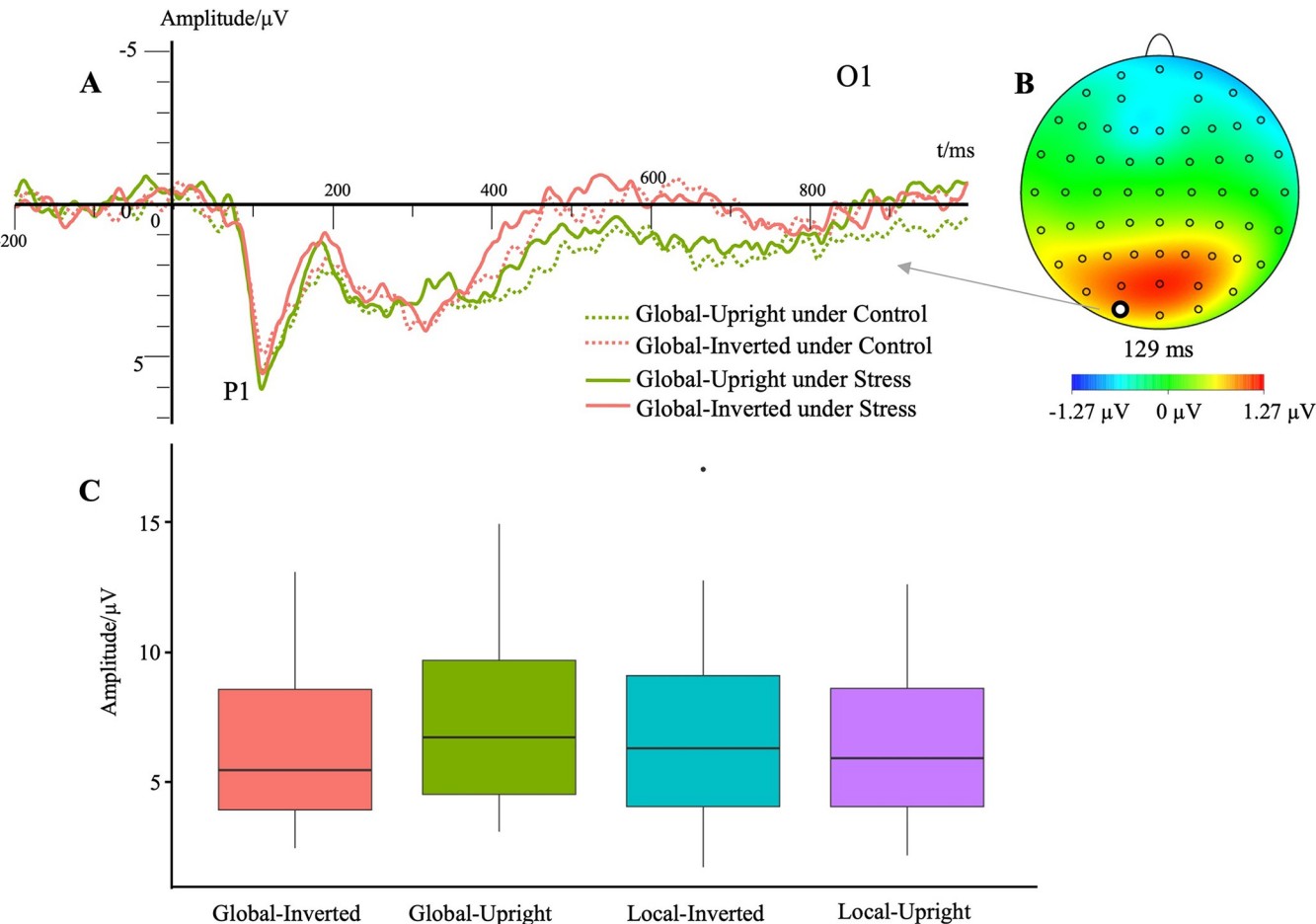

**Fig 4. Waveforms for the P1 component in different conditions.** (A) Each trace represents the grand average waveform of the P1 component at the electrode O1. The different colors illustrate the various experimental conditions. (B) The topographic map demonstrates the difference between global upright and global inverted. The red area indicates where the P1 component appears, and there is a meaningful difference between the two conditions. (C) P1 peak amplitudes for different conditions (averaged results among different stress levels and electrodes because of the non-significant effects).

**P1 peak amplitude.** The main effect of stress level was not significant, $F(1, 23) = 0.53$, $p > .05$, $\eta_p^2 = .02$. The main effect of contour characteristic was significant, $F(1,23) = 4.94$, $p < .05$, $\eta_p^2 = .18$, $M_{local} = 6.70$ μV, $M_{global} = 7.12$ μV. Post hoc testing showed that the P1 peak amplitude of global motion was significantly higher than that of local motion ($p < .05$). The main effect of motion characteristic was not significant, $F(1,23) = 1.75$, $p > .05$, $\eta_p^2 = .07$. The main effect of electrode site was not significant, $F(2.46,56.66) = 0.46$, $p > .05$, $\eta_p^2 = .02$. The interaction effect between the contour characteristic and motion characteristic was significant, $F(1,23) = 12.67$, $p < .01$, $\eta_p^2 = .36$. Simple effects analyses showed that P1 peak amplitude of inverted walking was significantly lower than that of upright walking in global motion ($p < .01$, **Fig 4A**). The P1 peak amplitude of global motion was significantly higher than that of local motion in the upright walking condition ($p < .01$).

**P2 peak latency.** The main effect of stress level was not significant, $F(1, 23) = 2.72$, $p > .05$, $\eta_p^2 = .11$. The main effect of contour characteristic was not significant, $F(1,23) = 3.02$, $p > .05$, $\eta_p^2 = .12$. The main effect of motion characteristic was significant, $F(1,23) = 23.07$, $p < .01$, $\eta_p^2 = .50$, $M_{upright} = 243.19$ ms, $M_{inverted} = 254.58$ ms. Post hoc testing showed that the P2 peak

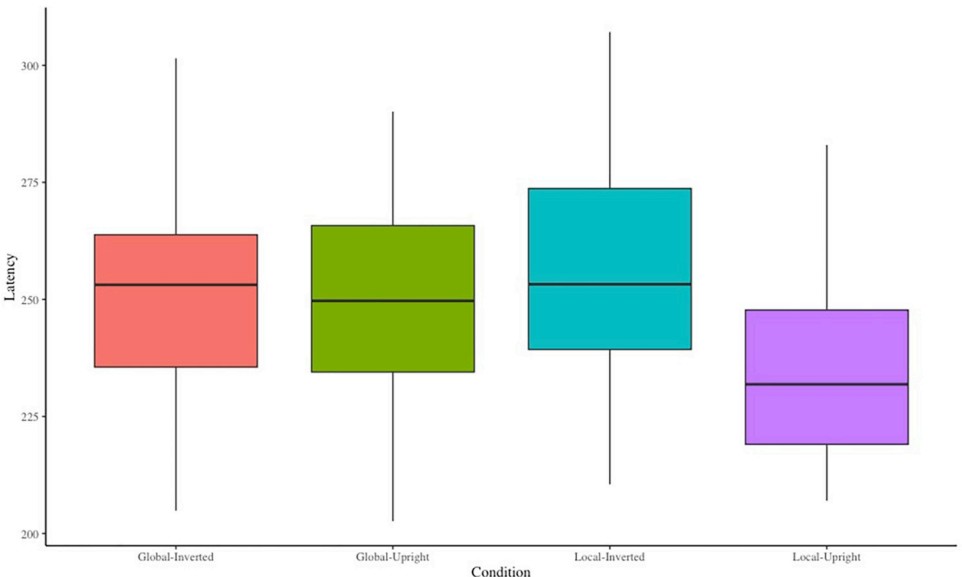

**Fig 5. P2 peak latency for each condition.** Please note that the illustrated data above are the averaged results of different stress levels and electrodes.

latency of upright walking was significantly shorter than that of inverted walking ($p < .01$). The main effect of the electrode site was not significant, $F(1.22,28.10) = 0.87$, $p > .05$, $\eta_p^2 = .04$. The interaction effect between motion characteristic and contour characteristic was significant, $F(1,23) = 12.83$, $p < .01$, $\eta_p^2 = .36$. Simple effects analyses showed that the P2 peak latency of inverted walking was significantly longer than that of upright walking in the local motion condition ($p < .01$, **Figs 5 and 6A**). The P2 peak latency of local motion was significantly shorter than that of global motion in the upright walking condition ($p < .01$, **Figs 5 and 6B**).

**P2 peak amplitude.** The main effect of stress level was significant, $F(1,23) = 8.84$, $p < .01$, $\eta_p^2 = .28$, $M_{stress} = 7.20$ μV, $M_{control} = 8.62$ μV. Post hoc testing showed that the P2 peak amplitude of the stress condition was significantly lower than that of the control condition ($p < .01$, **Fig 6A**). The main effect of contour characteristic was not significant, $F(1,23) = 0.40$, $p > .05$, $\eta_p^2 = .02$. The main effect of motion characteristic was significant, $F(1,23) = 22.24$, $p < .01$, $\eta_p^2 =$

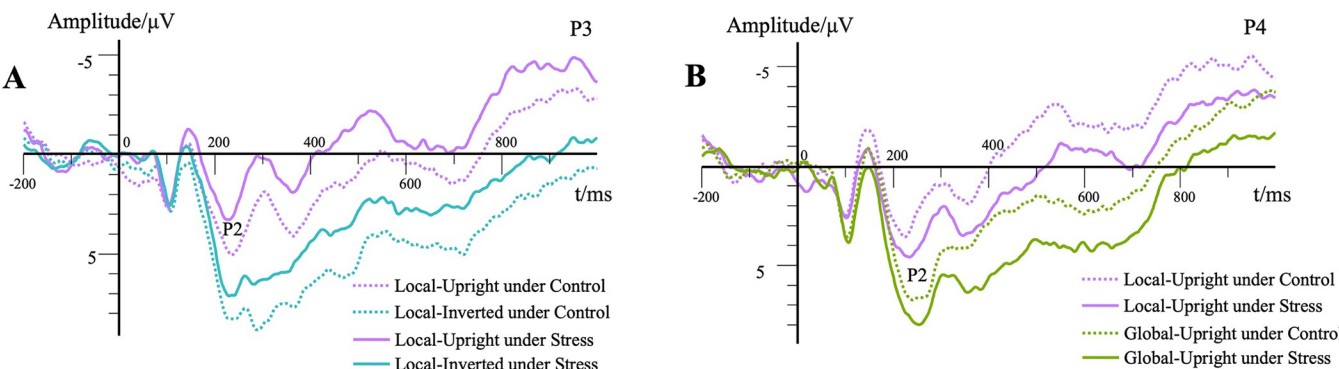

**Fig 6. Waveforms for the P2 component in different conditions.** (A) Each trace represents the grand average waveform of the P2 component for local motion conditions between different stress levels and motion characteristics at the electrode P3. (B) Each trace represents the grand average waveform of the P2 component for local motion conditions between different stress levels and contour characteristics at the electrode P4.

.49, M$_{upright}$ = 7.23 μV, M$_{inverted}$ = 8.60 μV. Post hoc testing showed that the P2 peak amplitude of inverted walking was significantly higher than that of upright walking ($p <$ .01). The main effect of the electrode site was significant, $F(1.51,34.76) = 3.82$, $p <$ .05, $\eta_p^2$ = .14. Post hoc testing showed that the P2 peak amplitude of the P3 site was significantly higher than that of the P5 site ($p <$ .01). The P2 peak amplitude of the P4 site was significantly higher than that of the P6 site ($p <$ .05). There was no significant difference between the other two sites ($p >$ .05).

The interaction effect between the electrode site and contour characteristic was significant, $F(2.15,49.48) = 10.31$, $p <$ .01, $\eta_p^2$ = .31. Simple effects analyses showed that the P2 peak amplitude of the P3 site was significantly higher than that of the P5 site in local motion or global motion conditions ($p <$ .01). The P2 peak amplitude of the P4 site was significantly higher than that of the P6 site in the global motion condition ($p <$ .01). The P2 peak amplitude of local motion was significantly higher than that of global motion at the P6 site condition ($p <$ .01). The interaction effects between stress level, electrode site, and motion characteristic were marginally significant, $F(1.58,36.33) = 3.40$, $p =$ .06, $\eta_p^2$ = .13. Simple effects analyses showed that P2 peak amplitude of inverted walking was significantly higher than that of upright walking regardless of electrode sites and stress levels ($p <$ .05). The P2 peak amplitude of stress condition was significantly lower than that of control condition regardless of electrode sites and motion characteristic ($p <$ .05, **Fig 6A**).

The interaction effect between motion characteristic and contour characteristic was significant, $F(1,23) = 82.96$, $p <$ .01, $\eta_p^2$ = .78. Simple effects analyses showed that the P2 peak amplitude of inverted walking was significantly higher than that of upright walking under the local motion condition ($p <$ .01). The P2 peak amplitude of inverted walking was significantly lower than that of upright walking under the global motion condition ($p <$ .01). The P2 peak amplitude of local motion was significantly higher than that of global motion under the inverted walking condition ($p <$ .01). The P2 peak amplitude of local motion was significantly lower than that of global motion under the upright walking condition ($p <$ .01). The interaction effect of electrode site, motion characteristic and contour characteristic was significant, $F(2.23,51.35) = 5.98$, $p <$ .01, $\eta_p^2$ = .21. The P2 waveforms for the global-upright conditions can be found in **Fig 7**.

**N330 peak latency.** The main effect of stress level was not significant, $F(1,23) = 0.10$, $p >$ .05, $\eta_p^2$ = .004. The main effect of contour characteristic was significant, $F(1,23) = 25.95$, $p <$ .01, $\eta_p^2$ = .53, M$_{local}$ = 302.26 ms, M$_{global}$ = 322.33 ms. Post hoc testing showed that the N330 peak latency of local motion was significantly shorter than that of global motion ($p <$ .01). The main effect of motion characteristic was significant, $F(1,23) = 27.76$, $p <$ .01, $\eta_p^2$ = .55, M$_{upright}$ = 299.55 ms, M$_{inverted}$ = 325.04 ms. Post hoc testing showed that the N330 peak latency of inverted walking was significantly longer than that of upright motion ($p <$ .01). The main effect of the electrode site was not significant, $F(1,23) = 0.84$, $p >$ .05, $\eta_p^2$ = .04. The interaction effect between electrode site and contour characteristic was significant, $F(1,23) = 5.23$, $p <$ .05, $\eta_p^2$ = .19. Simple effects analyses showed that N330 peak latency of local motion was significantly shorter than that of global motion at Fz or F1 electrode sites ($p <$ .01). The interaction effect between contour characteristic and motion characteristic was significant, $F(1,23) = 29.20$, $p <$ .01, $\eta_p^2$ = .56. Simple effects analyses showed that the N330 peak latency of inverted walking was significantly longer than that of upright walking under the global motion condition ($p <$ .01). The N330 peak latency of local motion was significantly shorter than that of global motion under the inverted walking condition ($p <$ .01). The N330 waveforms can be seen in the **Fig 8**.

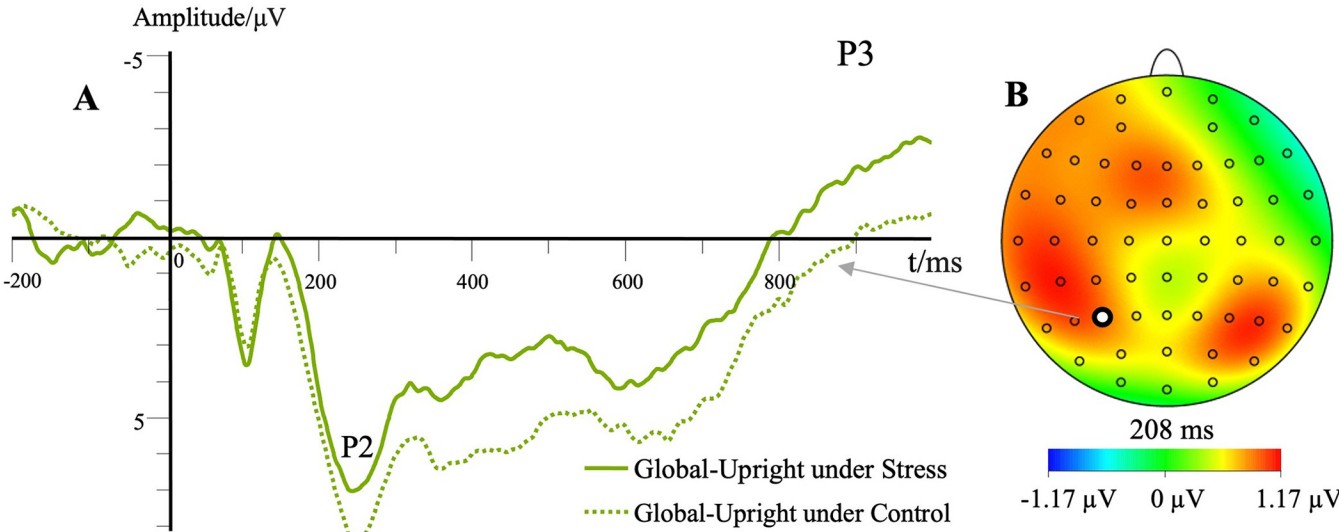

**Fig 7. Waveforms for the P2 component in different stress levels of the "global-upright" conditions.** (A) Each trace represents the grand average waveform of the P2 component at the electrode P3 (global upright). The different lines illustrate the different stress levels. The topographic map indicated the difference between the control and stress conditions. (B) The red area indicates where the P2 component appears, and there is a meaningful difference between the two conditions.

**N330 peak amplitude.** The main effect of stress level was significant, $F(1,23) = 9.35$, $p < .01$, $\eta_p^2 = .29$, $M_{stress}$ = -6.56 µV, $M_{control}$ = -4.14 µV. Post hoc testing showed that the N330 peak amplitude of biological motion perception under the stress condition was significantly higher than that under the control condition ($p < .01$, **Fig 9A**). The main effect of the contour characteristic was not significant, $F(1,23) = 0.17$, $p > .05$, $\eta_p^2 = .007$. The main effect of motion characteristic was not significant, $F(1,23) = 2.97$, $p > .05$, $\eta_p^2 = .11$. The main effect of electrode site was not significant, $F(1,23) = 0.54$, $p > 0.05$, $\eta_p^2 = .02$. The interaction effect between

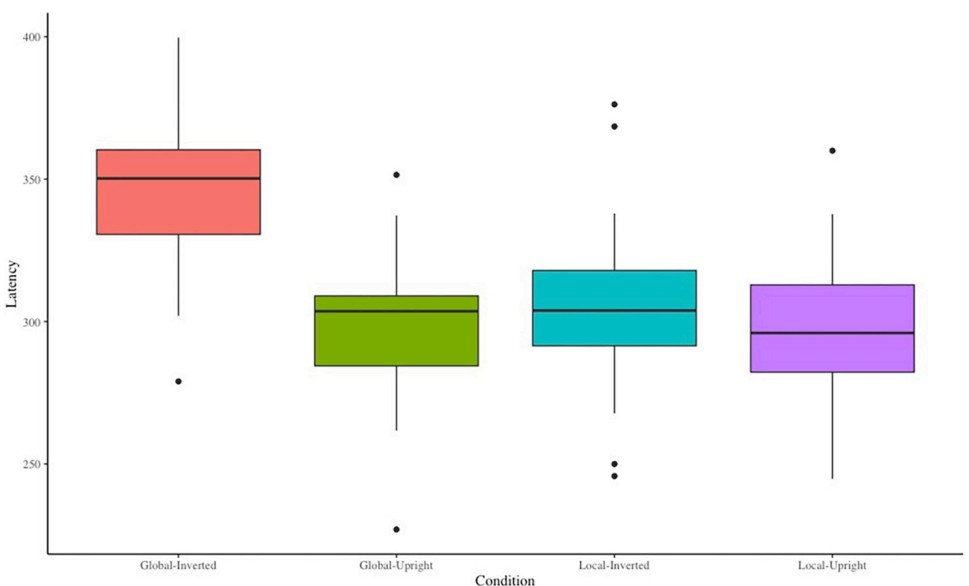

**Fig 8. N330 peak latency for each condition.**

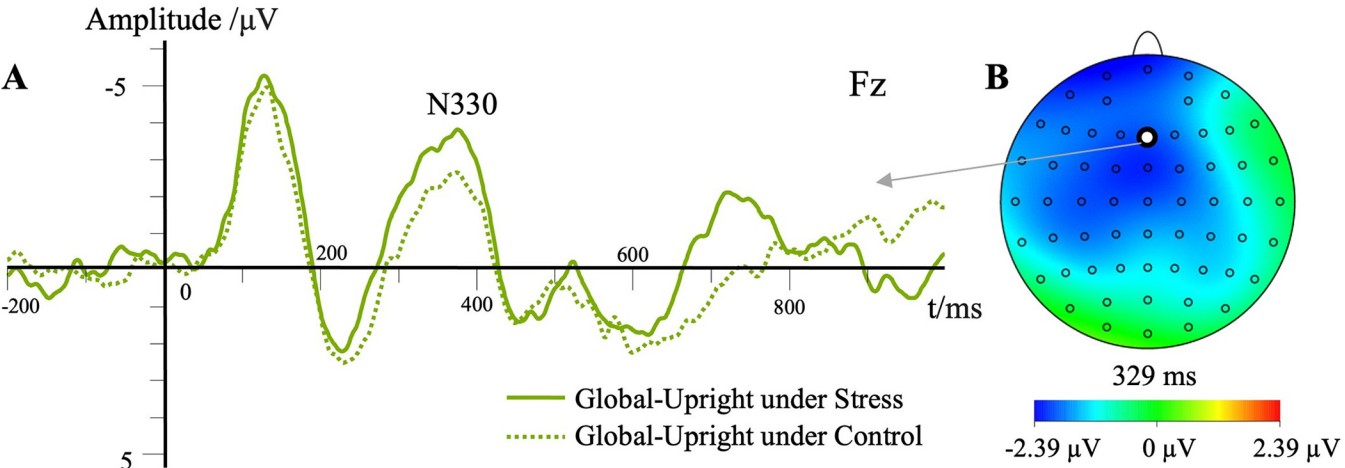

**Fig 9. Waveforms for the N330 component in different conditions.** (A) Each trace represents the grand average waveform of the N330 component at the electrode Fz (global upright). The different lines illustrate the different stress levels in the global upright condition. The topographic map indicated the difference between the stress and control conditions. (B) The blue area indicates where the N330 component appears.

contour characteristic and motion characteristic was significant, $F(1,23) = 12.11$, $p < .01$, $\eta_p^2 = .35$. Simple effects analyses showed that the N330 peak amplitude of inverted walking was significantly lower than that of upright walking under the local motion condition ($p < .01$). The N330 peak amplitude of local motion was significantly lower than that of global motion under the inverted walking condition ($p < .01$).

**LPP mean amplitude.** The main effect of stress level was significant, $F(1,23) = 13.68$, $p < .01$, $\eta_p^2 = .37$, $M_{stress} = 2.83$ μV, $M_{control} = 5.03$ μV. Post hoc testing showed that the LPP mean amplitude was significantly lower in the stress condition than in the control condition ($p < .01$, **Fig 10A**). The main effect of contour characteristic was significant, $F(1,23) = 26.58$, $p < .01$, $\eta_p^2 = .54$, $M_{local} = 3.10$ μV, $M_{global} = 4.77$ μV. Post hoc testing showed that the LPP mean amplitude was significantly higher in the global motion condition than in the local motion condition ($p < .01$). The main effect of motion characteristic was significant, $F(1, 23) = 4.60$, $p < .05$, $\eta_p^2 = .17$, $M_{upright} = 4.36$ μV, $M_{inverted} = 3.51$ μV. Post hoc testing showed that the LPP mean amplitude was significantly higher in the inverted walking condition than in the upright walking condition (p < 0.05). The main effect of the electrode site was significant, $F(2,46) = 7.46$, $p < .01$, $\eta_p^2 = .25$. Post hoc testing showed that LPP mean amplitude of biological motion perception at the CPz was significantly higher than that at the CP1 and CP2 ($p < .01$).

The interaction effect between electrode site and motion characteristic was significant, $F(1.6,36.44) = 7.05$, $p < .01$, $\eta_p^2 = .24$. The interaction effect between contour characteristic and motion characteristic was significant, $F(1,23) = 105.00$, $p < .01$, $\eta_p^2 = .82$. Simple effects analyses showed that the LPP mean amplitude of the inverted walking was significantly higher than that of the upright walking under the local motion condition ($p < .01$). The LPP mean amplitude of the inverted walking was significantly lower than that of the upright walking under the global motion condition ($p < .01$). The LPP mean amplitude of the local motion was significantly higher than that of the global motion under the inverted walking condition ($p < .01$). The LPP mean amplitude of the local motion was significantly lower than that of the global motion under the upright walking condition ($p < .01$). There was no significant interaction effect between two factors in the other experimental conditions ($p > .05$).

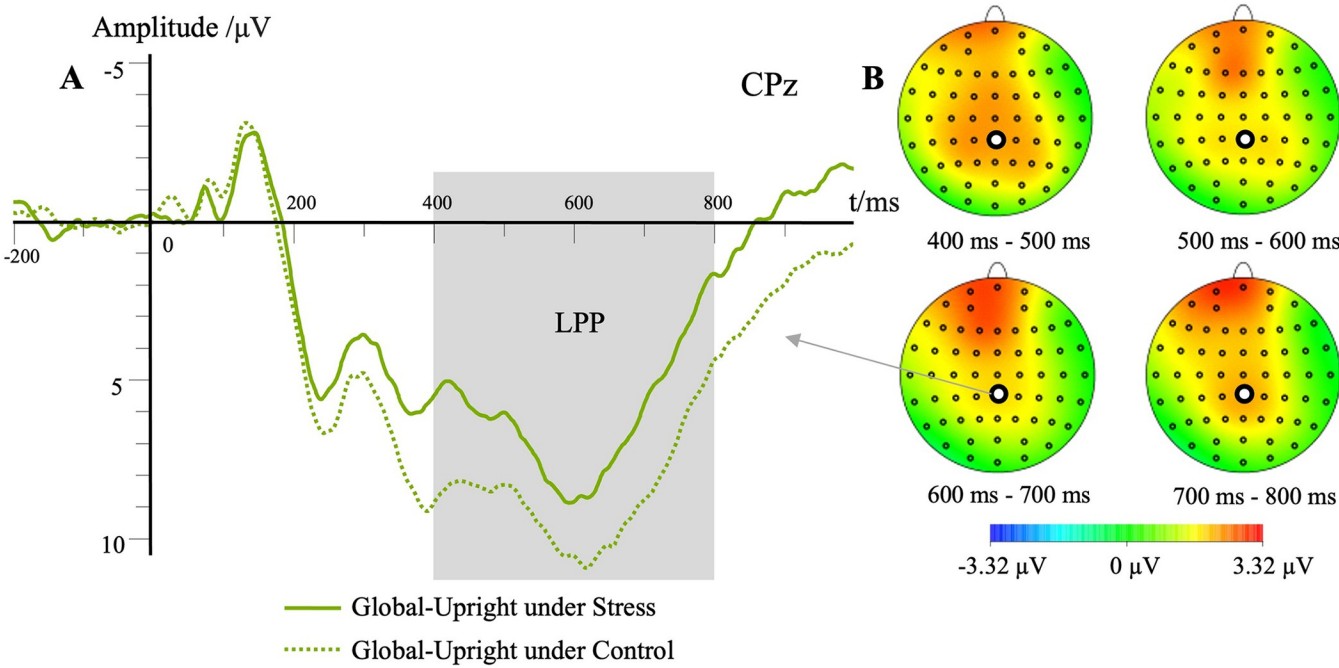

**Fig 10. Waveforms for the LPP in different conditions.** (A) Each trace represents the grand average waveform of LPP at the electrode CPz (global upright). The different lines illustrate the different stress levels in the global upright condition. The topographic map indicated the difference between the control and stress conditions. (B) The first-time segment between 400–500 ms showed the most significant differences between the two conditions.

## Discussion

This study provides empirical evidence that stress improves biological motion perception. RT was shorter, and attention control was better under stress conditions. There was a lower investment of attention resources in the early stage of biological motion perception processing. In the late stage, there was decreased inhibition persistence in the parietal-occipital region. The following sections will discuss these results.

The reaction time in the biological motion perception task was significantly shorter under the stress condition than under the control condition. A previous study found that there was shorter RT but lower accuracy of perceptual processing under the stress condition [40]. Other research showed that participants had a smaller attention range under the time pressure condition, which manifested as faster perceptual speed and lower accuracy [41], and that there was a faster perceptual speed under the stress condition due to the increase in alertness [4]. RT was shorter in the upright walking condition than in the inverted walking condition and was also shorter in the global motion condition than in the local motion condition. The accuracy was higher in the upright walking condition than in the inverted walking condition. These results indicated that there was an "inversion effect" in the process of biological motion perception, with longer RT when the set of light dots representing biological motion was inverted [42]. This inversion effect was induced by the disturbed structural information provided by the light spots [43]. Regardless of whether the information was about global or local motion, and regardless of whether the participants were under stress or not, there was a processing advantage in the upright walking condition.

P1 had shorter peak latency under the stress condition. P1 is the component associated with early visual information processing, which mainly involves the primary feature coding stage of visual information [20]. P1 latency is the time period in which visual information

processing occurs. Our results showed that P1 peak latency was longer in the inverted walking condition than in the upright walking condition, suggesting that people need to spend more time on pattern recognition when the motion stimulus is inverted. P1 peak latency was also longer in the global upright condition than in the local upright condition. Participants' selective attention was likely disrupted by the presentation of more cues (12 lights compared to four lights). P1 peak latency was longer at O1 and O2 than at POz electrode sites, indicating the extrastriate cortex of the central parieto-occipital area was the first to visually encode the moving dot pattern. The P1 peak amplitude was higher in the global upright motion condition than in the global inverted walking condition and the local upright walking condition. That is, compared with the other conditions, participants were more attracted to global upright walking. Finally, high arousal and alertness were induced by acute psychological stress [44]. Stress could make the sensory information input and early visual processing more sensitive [45], causing the P1 component to appear earlier.

The P2 peak latency was shorter in the local upright condition than in the local inverted and global upright conditions. This finding reflects P2's relationship with working memory skills. Working memory was required to complete the task, but working memory is also important for pattern recognition, in which external information is matched with the information stored in long-term memory [21]. Unlike local inverted walking, local upright walking does not require psychological rotation to process, making identification easier. The pattern recognition of local upright walking conforms to the normal action schema. However, matching speed may be slower because there are more dots to process in the representation of global upright walking than in the representation of local leg movement, and thus P2 appeared later. In addition, P2 peak amplitude was higher in the global upright and local inverted walking conditions than in the global inverted and local upright walking conditions, suggesting a higher allocation of attention in the perceptual judgment task [46, 47] in these conditions. The P2 peak amplitude of the P3 site was higher than that of the P5 site, suggesting that the Lingual Gyrus (LG) in the central occipital area was mainly involved in the pattern recognition of biological motion perception [9]. P2 peak amplitude was lower under the stress condition, indicating higher attention alertness and relatively lower attention resources during the task. That is, due to the influence of stress, the attention resources invested by people in the early stage of biological motion perception were relatively lower than in the control condition.

Stress had substantial effects on the reactivity of the N330 component, suggesting that participants needed to use more attention resources to process the biological motion perception tasks in a highly alert state. This result is consistent with the results of previous research showing that there was a state of high arousal and alertness induced by acute psychological stress [44]. In short, regardless of the motion characteristic and contour characteristic of biological motion, people are likely to show better attention control while perceiving body motion when highly alert. N330 peak amplitude was lower in the local inverted walking condition than in the local upright and global inverted walking conditions. A previous study found that the N330 component was induced by tasks requiring biological motion perception of upright and inverted walking [22]. In the current study, there were more light dots in the global inverted walking task than in the local inverted walking task, with the latter condition requiring less attention resources to identify the direction.

N330 had longer latency under the global inverted walking condition, suggesting that identifying global inverted walking was more difficult than identifying other forms of biological motion. There was reactivity of two negative components, namely N200 and N330, in the process of biological motion perception [48], and N330 is related to the attention process. The N300 peak amplitude was higher in the upright and inverted biological motion conditions than in messy motion [22]. N300 was closely related to the recognition of biological motion

patterns. Thus, the results of earlier research and our own research suggest that N330 is a specific negative component of biological motion perception and is related to the attention process and attention control ability.

The LPP had a lower mean amplitude under the stress condition. This result indicated that the inhibition persistence on biological motion perception tasks was weaker under the stress condition than under the control condition. There was a higher attention alertness and a narrower range of attention under stress. More attention resources were needed to cope with the stress, and thus, lower attention resources were required to process the perception of biological motion. The LPP was seen, as in earlier research, as a marker of top-down cognitive processing or the process of actively encoding external information [49]. The LPP component was activated in earlier research by the recognition of human activity and the process of biological motion perception [23]. In the current study, the LPP mean amplitude showed an increase in the process of biological motion perception. In prior research, LPP was used to assess inhibitory ability during visual cortex activity [50]. In addition, we found that the LPP mean amplitude was significantly higher in the global upright and local inverted walking conditions than in the local upright and global inverted walking conditions. The LPP was induced by complex discrimination tasks, such as determining the facing direction of a PLD walker [51]. We can conclude that inhibition persistence was longer in the global upright walking than in the local upright walking, suggesting the speed of pattern recognition was slower in the global upright walking condition than in the local upright walking condition. The inhibition persistence was longer in the local inverted walking than in the global inverted walking, indicating the direction of the PLD walker could be quickly judged by the moving track of the light dots simulating the arm's movement.

The behavioral results showed that the perceptual processing speed of biological motion perception was faster under the stress condition than under the control condition, and the EEG results showed shorter P1 peak latency and higher N330 peak amplitude under the stress condition than under the control condition. It can be inferred that the shorter RT in biological motion perception tasks under stress was related to the P2 and N330 components. A previous study found that psychological stress can make people more sensitive to the input of sensory information and early visual processing. Specifically, participants had a faster speed of perceptual processing under stress [40]. The ability of attention control has been shown to be improved under stress [52, 53].

In terms of practical significance, the results of this study suggest that acute psychological stress can promote performance on biological motion perception tasks. These results are consistent with the inverted U-shaped hypothesis [54], which is that a moderate level of psychological arousal is helpful to behavior performance in learning tasks. In terms of research limitations, this study used the PLD paradigm to simulate biological motion, but the stimuli are still very different from real human motion. In future research, we can consider using immersive virtual reality technology to study the characteristics of biological motion perception and to improve ecological validity in this line of research. In addition, the results of this study may be affected by stimulus-response compatibility (SRC), which refers to the consistency between the motion direction and the keystroke response. In future research, the keystrokes can be counterbalanced to avoid SRC as a possible confound.

## Conclusions

The current study showed that reaction time in the biological motion perception task was faster and the ability of attention control was improved under acute psychological stress. Under stress, attention resources were used earlier and were less invested in the early stage of

biological motion perception processing. In the late stage, the inhibition persistence of the parietal-occipital region was weakened under stress. In addition, there was an inversion effect in biological motion perception. This effect was related to the structural characteristics of biological motion perception but not to the state of acute stress.

## Supporting information

**S1 Appendix. Stimulus set about multiplication problems in the experiment.**
(DOCX)

**S2 Appendix. Key/answer mapping analysis for RT and accuracy.**
(DOCX)

**S3 Appendix. Key/answer mapping analysis for ERP.**
(DOCX)

## Acknowledgments

The authors would like to sincerely thank everyone who participated in the experiment for sharing their time. The authors also acknowledge support for the publication costs by the Open Access Publication Fund of Bielefeld University and the Deutsche Forschungsgemeinschaft (DFG).

## Author Contributions

**Conceptualization:** Jifu Wang, Lin Yu.

**Data curation:** Jifu Wang, Fang Shi.

**Formal analysis:** Jifu Wang, Lin Yu.

**Funding acquisition:** Jifu Wang, Lin Yu.

**Investigation:** Jifu Wang.

**Methodology:** Jifu Wang, Fang Shi, Lin Yu.

**Project administration:** Jifu Wang, Lin Yu.

**Supervision:** Jifu Wang, Fang Shi, Lin Yu.

**Writing – original draft:** Jifu Wang.

**Writing – review & editing:** Jifu Wang, Fang Shi, Lin Yu.

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
