## [Decision Letter · Decision Letter 0]

12 Jun 2024

PONE-D-23-34024Effects of Acute Stress on Biological Motion PerceptionPLOS ONE

Dear Dr. Yu,

Thank you for submitting your manuscript to PLOS ONE. After careful consideration, we feel that it has merit but does not fully meet PLOS ONE’s publication criteria as it currently stands. Therefore, we invite you to submit a revised version of the manuscript that addresses the points raised during the review process.

 Reviewers found value in the work but they raised several concerns that deserve to be carefully considered. In particular, both of them highlighted the need for a more clear presentation of the results, also from the point of view of figure production. Attention shoudl be devoted also in improving readability and soundness.

We look forward to receiving your revised manuscript.

Kind regards,

Alessandro Mengarelli

Academic Editor

PLOS ONE

 [Humanities and Social Sciences Projects Funded by the Ministry of Education and the Scientific Research Project of Hubei Provincial Department of Education, China (Grant No. Q20221313)].  

3. In the online submission form you indicate that your data is not available for proprietary reasons and have provided a contact point for accessing this data. Please note that your current contact point is a co-author on this manuscript. According to our Data Policy, the contact point must not be an author on the manuscript and must be an institutional contact, ideally not an individual. Please revise your data statement to a non-author institutional point of contact, such as a data access or ethics committee, and send this to us via return email. Please also include contact information for the third party organization, and please include the full citation of where the data can be found.

Additional Editor Comments (if provided):

Reviewers' comments:

Reviewer's Responses to Questions

**Comments to the Author**

1. Is the manuscript technically sound, and do the data support the conclusions?

Reviewer #1: Partly

Reviewer #2: Yes

2. Has the statistical analysis been performed appropriately and rigorously? 

Reviewer #1: Yes

Reviewer #2: Yes

3. Have the authors made all data underlying the findings in their manuscript fully available?

Reviewer #1: No

Reviewer #2: Yes

4. Is the manuscript presented in an intelligible fashion and written in standard English?

Reviewer #1: Yes

Reviewer #2: No

5. Review Comments to the Author

Reviewer #1: The concept of this study is interesting, but presentation of results is unclear. It would be better to explain the neural origin of each component (P1, P2, N330, especially LPP) and the significance of activity from each region (PO3, O1, POz, O2, PO4 for P1; P3 to P6 for P2; Fz and F1 for N330; CPz, CP1, and CP2 for LPP, also why the regions were selected) in the introduction, and then state what the authors wanted to clarify in this study. Grand average works well for checking the overall waveform, but is not suitable for statistical comparisons of each component. The amplitude and latency of each component of the ERP should be described by a box-and-whisker or bar graph. Since there appears to be more difference in amplitude than in latency for P1, the mean and standard deviation should be plotted to clarify the difference. In the results, the authors should explain why there is a difference between P3 and P5 sites in the peak amplitude of P2 component, and should also state what the difference means. Similarly, describe the behavioral data using figures.

The comparison of motion characteristic especially between “Global” and “Local” may have been influenced more by confusability than by the amount of information; if the Local figure were also placed so that the figure was centered rather than up and down, it would not be equivalent, but the confusability could be closer to that of the whole body. This is just a comment and please consider that next time. P1 should discuss which site caused the difference in latency and what its significance is. Since P1Peak is expressed at each site, we should discuss what this component indicates.

Minor comments

L239-240 It says that there is a significant difference in post-hoc P1 peak latency, but does not indicate which site.

Figure Legend does not indicate which the region is indicated.

Reviewer #2: The manuscript explores the impact of acute psychological stress on the perception of biological motion and investigates attention as a mediating factor for these effects. The researchers conducted an electrophysiological study to measure evoked responses during a 'bright moving dot' visual scene under conditions of induced stress.

This study is intriguing, but the manuscript is challenging to follow, largely due to problems with the English language.

These points are indicated below:

Line. 42, Rephrase the sentence. > “In the current study, we examined the effects of acute psychological stress on the perceptual processing of biological motion and attention as a mechanism of these effects”

Line 45, Two sentences redundant: “Acute stress can be induced by the improved Montreal Imaging Stress Task (MIST) [7]. The improved MIST task uses multiplication to induce acute stress.”

Line 67 Please rephrase “ In this study, the dynamic sequence diagram of biological light spots was used to study biological motion perception ability”

The volunteer college students, did they receive credit, or money?

“Inversion effect” is not clearly described

Line 150 please rephrase “The mental arithmetic items in the formal experiment (10 trials) were different from those in the formal experiment”.

Line 166, please modify the sentence “In the formal experiment, the red arrow will not be marked.”

Line 186, please rephrase “The length and bad interval free for ICA interval were both 50 s.”

Figure legends are not descriptive enough, they should be more comprehensive. This makes the figures hard to decipher.

Line 465, There needs a better explanation than the current for LPP: “We can conclude that inhibition persistence was longer in the global upright and local inverted walking conditions than in the local upright and global inverted walking conditions.”

6. PLOS authors have the option to publish the peer review history of their article (what does this mean?). If published, this will include your full peer review and any attached files.

Reviewer #1: No

Reviewer #2: **Yes: **Prof. Dr. Murat Özgören, MD PhD

---

## [Author Response · Author response to Decision Letter 0]

9 Aug 2024

Point 1: Please ensure that your manuscript meets PLOS ONE's style requirements, including those for file naming.

Response 1: Thanks for your suggestion. According to your suggestion, we have modified the manuscript to meet PLOS ONE’s style requirements.

Point 2: Thank you for stating the following financial disclosure: [Humanities and Social Sciences Projects Funded by the Ministry of Education and the Scientific Research Project of Hubei Provincial Department of Education, China (Grant No. Q20221313)].

Response 2: Thank you for the suggestion. According to your suggestion, we have modified in the section of “Funding information”. For details, see the red font on page 25, lines 854-859.

Point 3: In the online submission form you indicate that your data is not available for proprietary reasons and have provided a contact point for accessing this data.

Response 3: Thank you for the suggestion. According to your suggestion, we have modified in the section of “Support information”. For details, see the red font on page 26, lines 880-885.

Point 4: Please include a separate caption for each figure in your manuscript.

Response 4: Thanks for your suggestion. We have added a separate file for figure captions in our revised manuscript. 

Reviewer#1:

Point 1. The concept of this study is interesting, but presentation of results is unclear. It would be better to explain the neural origin of each component (P1, P2, N330, especially LPP) and the significance of activity from each region (PO3, O1, POz, O2, PO4 for P1; P3 to P6 for P2; Fz and F1 for N330; CPz, CP1, and CP2 for LPP, also why the regions were selected) in the introduction, and then state what the authors wanted to clarify in this study. 

Response 1: Thank you for your great suggestion. According to your suggestion, we have modified in the section of “Introduction”. For details, see the red font on page 3-4, lines 89-110.

Point 2. Grand average works well for checking the overall waveform, but is not suitable for statistical comparisons of each component. The amplitude and latency of each component of the ERP should be described by a box-and-whisker or bar graph. Since there appears to be more difference in amplitude than in latency for P1, the mean and standard deviation should be plotted to clarify the difference. 

Response 2: Thanks for your suggestion. We totally agree that statistical plots are also needed to illustrate the differences found in the tests. We added several boxplots for the reaction time result, as well as the latency results for the P1, P2, and N330. Please refer to the new plots.

Point 3. In the results, the authors should explain why there is a difference between P3 and P5 sites in the peak amplitude of P2 component, and should also state what the difference means. Similarly, describe the behavioral data using figures. 

Response 3: Thank you for your good suggestion. According to your suggestion, we have modified in the section of “Discussion”. For details, see the red font on page 22, lines 759-761. We also add a boxplot fir the reaction time results. Please refer to the new Fig.2 

Point 4. The comparison of motion characteristic especially between “Global” and “Local” may have been influenced more by confusability than by the amount of information; if the Local figure were also placed so that the figure was centered rather than up and down, it would not be equivalent, but the confusability could be closer to that of the whole body. This is just a comment and please consider that next time. P1 should discuss which site caused the difference in latency and what its significance is. Since P1 Peak is expressed at each site, we should discuss what this component indicates.

Response 4: Thank you for your good suggestion. According to your suggestion, we have modified in the section of “Discussion”. For details, see the red font on page 21, lines 739-738.

Minor comments

Point 1. L239-240 It says that there is a significant difference in post-hoc P1 peak latency, but does not indicate which site.

Response: Thanks for your suggestion. According to your suggestion, we have modified in the section of “Electrophysiological Data”. For details, see the red font on page 12, lines 365-366. 

Point 2. Figure Legend does not indicate which the region is indicated.

Response 2: Thanks for your suggestion. We re-arranged the electrode label (legend) in the plots and marked the electrodes on the topographic map and the captions.

Reviewer #2: 

Point 1. Line. 42, Rephrase the sentence. “In the current study, we examined the effects of acute psychological stress on the perceptual processing of biological motion and attention as a mechanism of these effects”

Response 1: Thank you for your good suggestion. According to your suggestion, we have modified on the page 2, line 55.

Point 2. Line 45, Two sentences redundant: “Acute stress can be induced by the improved Montreal Imaging Stress Task (MIST) [7]. The improved MIST task uses multiplication to induce acute stress.”

Response 2: Thank you for your good suggestion. According to your suggestion, we have modified on the page 2, line 59.

Point 3. Line 67 Please rephrase “In this study, the dynamic sequence diagram of biological light spots was used to study biological motion perception ability”

Response 3: Thank you for your good suggestion. According to your suggestion, we have modified on the page 3, line 86.

Point 4. The volunteer college students, did they receive credit, or money? “Inversion effect” is not clearly described.

Response 4: Thank you for your good suggestion. According to your suggestion, we have modified on the page 5, lines 156-158 and page 1, lines 17-18.

Point 5. Line 150 please rephrase “The mental arithmetic items in the formal experiment (10 trials) were different from those in the formal experiment”.

Response 5: Thank you for your good suggestion. According to your suggestion, we have modified on the page 7, line 209.

Point 6. Line 166, please modify the sentence “In the formal experiment, the red arrow will not be marked.”

Response 6: Thank you for your good suggestion. According to your suggestion, we have modified on the page 8, lines 236-237.

Point 7. Line 186, please rephrase “The length and bad interval free for ICA interval were both 50 s.”

Response 7: Thank you for your good suggestion. According to your suggestion, we have modified on the page 9, line 303.

Point 8. Figure legends are not descriptive enough, they should be more comprehensive. This makes the figures hard to decipher.

Response 8: Thank you for your good suggestion. According to your suggestion, we have modified from Fig.1 to Fig.8. The descriptive content was added in the figure legends.

Point 9. Line 465, There needs a better explanation than the current for LPP: “We can conclude that inhibition persistence was longer in the global upright and local inverted walking conditions than in the local upright and global inverted walking conditions.”

Response 9: Thank you for your good suggestion. According to your suggestion, we have modified on the pages 23-24, lines 803-819.

---

## [Decision Letter · Decision Letter 1]

2 Sep 2024

Effects of acute stress on biological motion perception

PONE-D-23-34024R1

Dear Dr. Yu,

We’re pleased to inform you that your manuscript has been judged scientifically suitable for publication and will be formally accepted for publication once it meets all outstanding technical requirements.

Kind regards,

Alessandro Mengarelli

Academic Editor

PLOS ONE

Additional Editor Comments (optional):

All the Reviewers' concerns have been addressed in a proper way, and the paper is suitable for being published in its present form.

Reviewers' comments:

Reviewer's Responses to Questions

**Comments to the Author**

1. If the authors have adequately addressed your comments raised in a previous round of review and you feel that this manuscript is now acceptable for publication, you may indicate that here to bypass the “Comments to the Author” section, enter your conflict of interest statement in the “Confidential to Editor” section, and submit your "Accept" recommendation.

Reviewer #1: All comments have been addressed

Reviewer #2: All comments have been addressed

2. Is the manuscript technically sound, and do the data support the conclusions?

Reviewer #1: Yes

Reviewer #2: Yes

3. Has the statistical analysis been performed appropriately and rigorously? 

Reviewer #1: Yes

Reviewer #2: Yes

4. Have the authors made all data underlying the findings in their manuscript fully available?

Reviewer #1: Yes

Reviewer #2: Yes

5. Is the manuscript presented in an intelligible fashion and written in standard English?

Reviewer #1: Yes

Reviewer #2: Yes

6. Review Comments to the Author

Reviewer #1: Thank you for replying to my comments. The addition of the figure clarifies the difference. Very nice work. I have no further comments.

Reviewer #2: Even though the English is hard to follow the ms has improved substantially. The authors responded to the reviwers comments.

7. PLOS authors have the option to publish the peer review history of their article (what does this mean?). If published, this will include your full peer review and any attached files.

Reviewer #1: No

Reviewer #2: No

---

## [Editor Report · Acceptance letter]

9 Sep 2024

PONE-D-23-34024R1 

PLOS ONE

Dear Dr. Yu, 

I'm pleased to inform you that your manuscript has been deemed suitable for publication in PLOS ONE. Congratulations! Your manuscript is now being handed over to our production team.

Kind regards, 

on behalf of

Dr. Alessandro Mengarelli 

Academic Editor

PLOS ONE